# The Real-World Efficacy and Safety of Direct-Acting Antivirals for Chronic Hepatitis C in Patients Active Malignancies [note 1]

**DOI:** 10.3390/cancers16173114

**Published:** 2024-09-09

**Authors:** Maria Dąbrowska, Jerzy Jaroszewicz, Marek Sitko, Justyna Janocha-Litwin, Dorota Zarębska-Michaluk, Ewa Janczewska, Beata Lorenc, Magdalena Tudrujek-Zdunek, Anna Parfieniuk-Kowerda, Jakub Klapaczyński, Hanna Berak, Łukasz Socha, Beata Dobracka, Dorota Dybowska, Włodzimierz Mazur, Łukasz Ważny, Robert Flisiak

**Affiliations:** 1Department of Infectious Diseases and Hepatology, Medical University of Silesia, 40-635 Katowice, Poland; 2Department of Infectious and Tropical Diseases, Jagiellonian University, 30-688 Kraków, Poland; 3Department of Infectious Diseases and Hepatology, Medical University Wroclaw, 51-149 Wroclaw, Poland; justynajanocha@o2.pl; 4Department of Infectious Diseases and Allergology, Jan Kochanowski University, 25-317 Kielce, Poland; dorota1010@tlen.pl; 5Department of Basic Medical Sciences, Faculty of Health Sciences, Medical University of Silesia, 41-902 Bytom, Poland; 6Pomeranian Center of Infectious Diseases, Department of Infectious Diseases, Medical University of Gdańsk, 80-214 Gdańsk, Poland; 7Department of Infectious Diseases and Hepatology, Medical University of Lublin, 20-081 Lublin, Poland; 8Department of Infectious Diseases and Hepatology, Medical University of Białystok, 15-569 Białystok, Polandrobert.flisiak1@gmail.com (R.F.); 9Department of Internal Medicine and Hepatology, The National Institute of Medicine of the Ministry of Interior and Administration, 02-507 Warszawa, Poland; 10Daily Department of Hospital for Infectious Diseases in Warsaw, 01-201 Warszawa, Poland; 11Department of Infectious Diseases, Hepatology and Liver Transplantation, Pomeranian Medical University, 71-455 Szczecin, Poland; 12MedicalSpec Medical Center, 53-228 Wrocław, Poland; 13Department of Infectious Diseases and Hepatology, Faculty of Medicine, Collegium Medicum Bydgoszcz, Nicolaus Copernicus University, 87-100 Toruń, Poland; d.dybowska@wsoz.pl; 14Clinical Department of Infectious Diseases, Specialist Hospital in Chorzów, Medical University of Silesia, 41-500 Katowice, Poland

**Keywords:** direct-acting antivirals, hepatitis C virus, hepatocellular carcinoma, hematological disease, solid malignant tumor, treatment efficacy, treatment safety

## Abstract

**Simple Summary:**

In the era of direct-acting antiviral (DAA) agents, chronic hepatitis C virus (HCV) infection has become a curable disease. Eradication of the virus remains a major goal for the World Health Organization (WHO) by 2030. Main obstacles seem to be the lack of national screenings and shortage of knowledge among patients and healthcare professionals. There also remain, scarcely described in the literature, specific groups of patients who, due to their comorbidities such as malignant tumors, may not be considered as candidates eligible for DAA treatment. In our study, we aimed to characterize and present treatment efficacy in individuals with chronic hepatitis C and an active malignancy treated in Poland in years 2015–2020 with DAAs and compare their outcomes with a treated population with no active malignancy. The obtained results indicate high effectiveness and a low number of premature treatment discontinuations for the majority of patients with active malignancies, with some concerns around HCCs. We believe that data provided by this study will lead to more efficient elaboration of the standard of care in this population.

**Abstract:**

Background: Over the past years, the introduction of direct-acting antivirals (DAAs) revolutionized chronic hepatitis C treatment. We aimed to characterize and assess treatment efficacy in three specific groups of patients treated with DAAs: those with active solid malignant tumors (SMTs), hematological diseases (HDs) and hepatocellular carcinomas (HCCs). Methods: A total of 203 patients with active oncological disease (SMT *n* = 61, HD = 67, HCC *n* = 74) during DAA treatment in 2015–2020 selected from the EpiTer-2 database were analyzed retrospectively and compared to 12,983 patients without any active malignancy. Results: Extrahepatic symptoms were more frequent in HD patients (17.2% vs. SMT = 10.3%, HCC = 8.2%, without = 7.8%, *p* = 0.004). HCC patients characterized with the highest ALT activity (81 IU/L vs. SMT = 59.5 IU/L, HD = 52 IU/L, without = 58 IU/L, *p* = 0.001) more often had F4 fibrosis as well (86.11% vs. SMT = 23.3%, HD = 28.8%, controls = 24.4%, *p* = 0.001). A significant majority of subjects in HCC, HD and SMT populations completed the full treatment plan (HCC = 91%; *n* = 67, HD = 97%; *n* = 65, SMT = 100%; *n* = 62). Concerning the treatment efficacy, the overall sustained virologic response, excluding non-virologic failures, was reported in 93.6% HD, 90.16% SMT and 80.6% in HCC patients. Conclusions: As presented in our study, DAA therapy has proven to be highly effective and safe in patients with active SMTs and HDs. However, therapy discontinuations resulting from liver disease progression remain to be the major concern in HCC patients.

## 1. Introduction

The implementation of direct-acting antivirals (DAAs) targeting hepatitis C virus (HCV) at the beginning of the last decade has revolutionized treatment of HCV infections. Due to DAAs’ high efficacy in virus clearance [1], the medication facilitates an opportunity to eradicate HCV globally. This remains a major challenge for the World Health Organization (WHO) by 2030 [2].

This issue still needs to be addressed in Poland, as according to Polish Group of Experts for HCV, 2019–2023 have brought about a systematic decrease in the number of treated patients. The reasons for this deterioration seem to be complex, but above all, the influence of COVID-19 pandemics, unawareness among patients and physicians and lack of national screening program may be the main obstacles [3]. 

However, there are specific populations of patients, which due to underlying conditions such as an active oncological disease, may require special consideration while administering DAAs. Major concerns may include drug-to-drug interactions, treatment safety and time required to obtain sustained virological response (SVR). 

On the other hand, liver metastases or anti-cancer agents’ toxicity may as well substantially deteriorate liver function, which poses another threat to survival and the quality of life in patients with chronic hepatitis C. Obviously, potential benefits from the eradication of HCV in this population are much desired. Not only would efficient chronic hepatitis C treatment improve the overall liver function in these patients but it might also lead to improved outcomes in their comorbidities and most importantly, enhance successful anti-cancer therapy. The prevention of liver damage obtained by SVR would decrease the complication rates of a chronic condition such as cancer and the mortality rate resulting from the decompensation of a cirrhotic liver and a hepatocellular carcinoma. 

This has been clearly stated in a paper by Kamp WM et al., which concluded that DAAs improve overall survival in patients with hepatitis C and HCCs [4]. Moreover, a case series of 30 patients treated concomitantly with DAAs and anticancer agents (such as tamoxifen, platinum derivates and paclitaxel) described by F. Patauner et al. concludes with entirely successful and safe antiviral treatments in the individuals [5]. 

Therefore, it is crucial that potential chronic HCV infections be consistently detected and treated in patients suffering from HCCs, SMTs or HDs. This might require elaboration of a novel systemic approach in the overall cancer treatment planning.

Data on DAA treatment in these groups are scarce, which may result in the limited administration of highly effective anti-HCV medication in oncological patients.

This paper aims to highlight the efficacy and safety of DAA therapy in populations suffering from both chronic hepatitis C and active malignancies, namely those with SMTs, HDs and HCCs. The analysis includes characteristics, treatment specifics and outcomes of the three extracted groups of patients in comparison to those without the diagnosis of active oncological diseases during DAA treatment in real-life settings. 

## 2. Materials and Methods

A retrospective study was conducted using a real-life multi-center national database. EpiTer-2 is an ongoing study, supported by the Polish Association of Epidemiologists and Infectiologists. The study gathers data on patients with chronic hepatitis C treated from 2015 with DAAs in 22 hepatology centers in Poland. Demographic, clinical and antiviral treatment specifics were completed retrospectively and submitted through a web-based platform following the National General Data Protection Regulation in Poland. Informed consent was collected from patients before the start of the treatment due to the requirements of the National Health Fund. The data collected initially were not aimed at scientific research but rather at assessing the effectiveness and safety of registered medications in real-world clinical environments. The patients did not receive any experimental treatments. According to the local law in effect during this study (the Polish Pharmaceutical Law of 6 September 2001, art. 37al), non-interventional studies did not require approval from an ethics committee. The baseline information gathered included demographic and clinical data: age, gender, body mass index (BMI), comorbidities and concurrent medications, severity of liver disease and coinfections with hepatitis B virus (HBV) and human immunodeficiency virus (HIV), as well as the antiviral regimen previously utilized.

The grade of liver disease was evaluated noninvasively by the transient elastography (TE) or shear-wave elastography (SWE) or histologically by the liver biopsy. Assigning patients to the liver fibrosis stage ranging from F0 to F4 was based on the degree of the liver stiffness according to the METAVIR score from the European Association for the Study of the Liver (EASL) [2]. Patients with cirrhosis were evaluated in the Child–Pugh (CP) scale and the Model of End-Stage Liver Disease (MELD) scale, and the presence of esophageal varices was identified. 

Baseline characteristics of the patients at the onset of therapy included laboratory data such as serum alanine transaminase (ALT) levels, bilirubin concentration, albumin, creatinine, hemoglobin, white blood cell (WBC) counts and platelet counts, along with HCV viral load. HCV RNA was assessed at the beginning and end of treatment, as well as at least 12 weeks post-therapy. The evaluation of viral load was conducted using real-time polymerase chain reaction assays, while genotyping was performed through reverse hybridization assays.

The fundamental efficacy outcome was sustained virologic response (SVR) defined as a negative result of HCV RNA 12 weeks after treatment. The intention-to-treat group (ITT) included patients who received at least one dose of antiviral drug, and the per-protocol group (PP) was established by excluding patients because of non-virologic failure.

In the current analysis, 203 patients with reported malignancies compared to 12,983 patients with no active malignancy (NAM) treated in real-life settings were enrolled. Patients suffering from three main types of malignancies were extracted, HCCs (*n* = 74), SMTs (*n* = 62) and HDs (*n* = 67), as shown in Figure 1.

Results are expressed as mean ± standard deviation (SD), median and interquartile range (IQR). To compare quantitative variables researchers used the Kruskal–Wallis or ANOVA test for continuous variables. The qualitative variables were compared using the Pearson chi-square test. A *p*-value of less than 0.05 was considered significant. The analysis was performed in the R language in the RStudio environment.

## 3. Results

### 3.1. Characteristics of the Study Population

A total of 203 patients with reported active malignancies were enrolled in the analysis. The proportion of those patients among chronic hepatitis C patients was lower than that of the incidence in the general population of Poland (any malignancy 1.5% vs. 3.1%; HD 0.26% vs. 0.49%) [6]. The median age of the patients with malignancies was significantly higher in comparison to those without (HCC 62; HD 59; SMT 62 vs. NAM 52 years). On the contrary, median BMI level was almost even in all studied populations. The rate of comorbidities was substantially higher in the patients suffering from HCC (66.2% vs. HD 36.3%, SMT 43.7%, NAM 38.3%) and specifically arterial hypertension as well as diabetes were more prevalent in HCC patients (Table 1).

The distribution of genotypes was comparable in all analyzed groups, with the most predominant genotype being 1b. HCV viral load was notably higher in hemato-oncologic patients, with values of 6.2 log10 IU/mL for HD, 5.7 for SMT and 5.8 for HCC (*p* < 0.001). Additionally, the incidence of extrahepatic symptoms was greater in HD patients, recorded at 17.2%, compared to 10.3% for SMT, 8.2% for HCC and 7.8% for controls (*p* = 0.04). Moreover, patients with HCCs exhibited elevated ALT activity at 81 IU/L, while SMT and controls had 58 IU/L, and HD had 52 IU/L (*p* = 0.001). Patients with HCCs also presented a higher prevalence of F4 fibrosis at 83.8%, contrasted with 22.6% for SMT, 28.3% for HD and 24.0% for controls (*p* = 0.001).

Detailed characteristics regarding the severity of liver disease depending on the presence of malignancy are presented in Table 2.

### 3.2. Efficacy

Sustained virologic response rates were observed at 89.55% for HD, 90.32% for SMT and 77.03% for HCC in the Intend to Treat (ITT) analysis. After excluding non-virologic failures in the Per Protocol (PP) analysis, the rates were 93.6% for HD, 90.16% for SMT and 80.6% for HCC (Table 3 and Figure 2).

### 3.3. Safety

Among patients with HCCs, 67 (90.6%) underwent DAA therapy according to schedule, while seven (9.4%) discontinued prematurely (Table 4). 

In the group of subjects with hematological disease, 65 individuals (97%) underwent full treatment and two (3%) discontinued (Table 5).

On the contrary, all (*n* = 62; 100%) of the patients with active solid malignancy completed full schedule of therapy (Figure 3).

Finally, during DAA therapy and in the 12 weeks of follow-up, 69 deaths among the entire analyzed population were observed. Among them, 12 individuals (17.4%) had HCCs, two (2.9%) STMs and two (2.9%) hematologic malignancy.

## 4. Discussion

The introduction of new antiviral drugs targeting HCV at the beginning of the last decade marked a significant breakthrough in the treatment of HCV infections [1,7]. Prior to this, therapies based on interferon and ribavirin were characterized by low effectiveness and a wide range of side effects. The new drugs, referred to as direct-acting antiviral agents (DAAs), directly inhibit specific viral proteins, including the NS3 protease, NS5A and NS5B polymerase, effectively blocking the viral replication cycle and achieving an efficacy rate of over 90% (measured as SVR) [8]. Despite the impressive results of DAAs in eliminating the virus from the general population of individuals infected with HCV, it is also essential to monitor their effectiveness in special populations, particularly those with malignancies. Oncological patients undergoing anticancer treatment, including chemotherapy, are a particularly vulnerable group in the context of DAA treatment, as their condition often involves immunosuppression and potential interactions between anticancer treatment and antiviral therapy can occur [9].

This study aimed to evaluate the efficacy and safety of direct-acting antiviral (DAA) therapy in chronic hepatitis C patients with concurrent oncological conditions, including hepatocellular carcinomas (HCCs), solid malignant tumors (SMTs) and hematological diseases (HDs). The results provide critical insights.

The efficacy of direct-acting antiviral (DAA) therapy was evident across all patient groups, although differences emerged when comparing patients with and without malignancies. Patients with HDs and SMTs showed high rates of sustained virological response (SVR), at 89.55% and 90.32% respectively (ITT analysis), which is comparable to the general population without malignancies. However, the SVR rate for patients with HCCs was significantly lower at 77.03% (ITT analysis) and 80.6% (PP analysis). This suggests that while DAA therapy is effective in oncological populations, patients with HCC might experience reduced efficacy possibly due to advanced liver disease or tumor-related factors. These results are consistent with the literature data. A meta-analysis by Ji et al. indicated a lower SVR rate in patients treated with DAAs who have HCCs compared to those without HCCs (89.6%, 95% CI 86.8–92.1% vs. 93.3%, 95% CI 91.9–94.7%, *p* = 0.0012) [10]. Furthermore, a meta-regression analysis conducted by Ji et al. showed that HCC patients had a 4.8% reduction in SVR rate (95% CI 0.2–7.4%) compared to non-HCC patients [10]. However, SVR rates for HCC patients on DAA treatment varied significantly across the 49 studies included in this meta-analysis, ranging from 62.1% (Cheung et al., 2016) [10] to 100% (Barone et al., 2018) [11,12].

The safety profiles demonstrated a noteworthy high completion rate of DAA therapy among SMT patients, amounting to 100%. Additionally, the discontinuation rates were low for the HD (3%) and HCC (9.4%) populations. It is important to highlight that the mortality rates during DAA therapy and within 12 weeks of follow-up were higher for the HCC patient population (17.4%), which emphasizes the severe prognosis associated with advanced liver disease and cancer. This underscores the need for close monitoring and potentially more personalized therapeutic approaches for HCC patients. Future studies could expand on this research by obtaining data on HCC recurrence in the years following the conclusion of DAA therapy. It is worth noting that some previous studies have reported potential negative effects of DAAs on HCC recurrence [13], but subsequent studies have shown no correlation and there are currently no indications for withholding DAAs in HCC patients who respond well to the treatment [14,15]. 

Furthermore, considering the natural course of HCV infection and DAA safety among patients with malignancies, it is crucial to monitor individuals with ALT elevation indicating acute liver failure [16]. This complication is an uncommon one [17,18,19] but may require anticancer treatment regimens adjustment or halting [20]. ALT flares and exacerbations of HCV have several risk factors such as genotype 2 infection [21,22], the use of higher doses of corticosteroids and rituximab [20]. What is more, the risk of HCV acute exacerbation in novel anticancer therapies such as immune checkpoint inhibitors (e.g., nivolumab, durvalumab and pembrolizumab) or chimeric antigen receptor (CAR) T-cell therapy is not well-defined [23,24]. In general, achieving SVR through DAA treatment can lead to improved liver function and help prevent acute exacerbations of HCV as well as ALT flares during and after chemotherapy for cancer in the short term. In the long term, SVR may prevent the development of fibrosing sclerosing hepatitis or cirrhosis, which in turn could help avoid liver failure and HCCs [25,26]. HCV screening should be encouraged among cancer patients in order to ultimately enhance the prognosis for the infected. Chances of HCV eradication should not be avoided [27]. Our study lacks in a prospective presentation of ALT levels in SMT, HD or HCC patients. There were three reported liver disease decompensations in HCC group. This corresponds to the infrequent incidence of this complication in the general population of HCV infected. Additionally, we do not have data considering specific anticancer treatment schemes administered in observed patients, which could be an interesting insight into safety and potential drug-to-drug interactions. 

This study demonstrated that the HCC patients exhibited a greater prevalence of comorbidities, such as arterial hypertension and diabetes, compared to other patient groups. This highlighted that the burden of comorbidities in HCC patients likely contributes to the lower SVR rates observed and emphasizes the need for integrated management approaches that address both oncological and hepatic conditions simultaneously. As previously mentioned, lower SVR rates in HCC patients on DAA treatment have been confirmed in previous publications [11]. Some studies have suggested that these results may be related to the existence of HCV reservoirs in HCC cells and limited blood access to cancer cells, which may hinder the penetration of DAAs into these sites [15].

The severity of liver disease was significantly higher in HCC patients, with 86.11% presenting F4 fibrosis compared to 23.3% in SMT and 28.8% in HD patients. Additionally, higher ALT levels and liver stiffness measurements further corroborated the advanced liver disease in HCC patients. This advanced liver disease may affect patients’ response to DAA therapy and overall prognosis, emphasizing the importance of considering liver disease severity when planning HCV treatment in oncological settings. 

Although in current study, liver stiffness has only been measured at baseline, it is crucial to underline that it seems to be a compelling parameter for further investigation. In a recent observational five-year cohort study on 153 individuals undergoing anti-HCV treatment by Facciorusso A. et al., the authors indicate that liver stiffness declines significantly after achieving SVR; however, the effect seems to be the strongest in the first year after treatment completion [28]. Moreover, an observational and prospective study by Badia Aranda E. et al. involving multiple laboratories and elastography tests in 169 patients cured with DAAs, concluded with a documented improvement of investigated parameters, although the risk of liver function decompensation or HCC was remaining [29]. 

This study reported on a comparable distribution of HCV genotypes between patients with and without malignancies. However, a significantly higher viral load was observed in HD patients, which might indicate a more aggressive viral replication in this group. Despite this, HD patients achieved high SVR rates, indicating the robustness of DAA therapy even in high viral load scenarios.

There are certain limitations of current study that should be taken into account while interpreting the results. One of them seems to be its retrospective and observational design, which may result in lack of some important data such as treatment-adherence records or adverse events data that may be underreported in real-world settings. Furthermore, another limitation are the relatively small sample sizes of HCC, HD and SMT groups. 

However, presented analysis is one of very few characteristics representing oncological patients among those treated for chronic hepatitis C. This multi-center analysis covers a heterogeneous population as a solid representation of the real-world population in the clinical environment.

## 5. Conclusions

In summary, the findings of this research clearly show the significant efficacy of DAA therapy in patients with active malignancy, especially. The main factor contributing to lower SVR in patients with HCCs is early mortality due the disease progression and high mortality during the course of treatment. This study also underscores the urgent need for better HCC surveillance in chronic hepatitis C patients. 

## Figures and Tables

**Figure 1 cancers-16-03114-f001:**
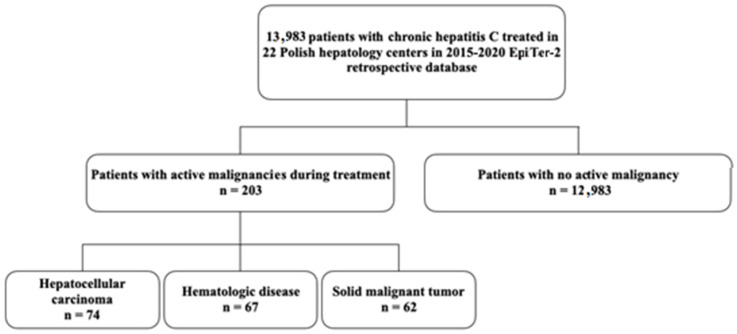
Flow chart showing the selection of patients included in this study.

**Figure 2 cancers-16-03114-f002:**
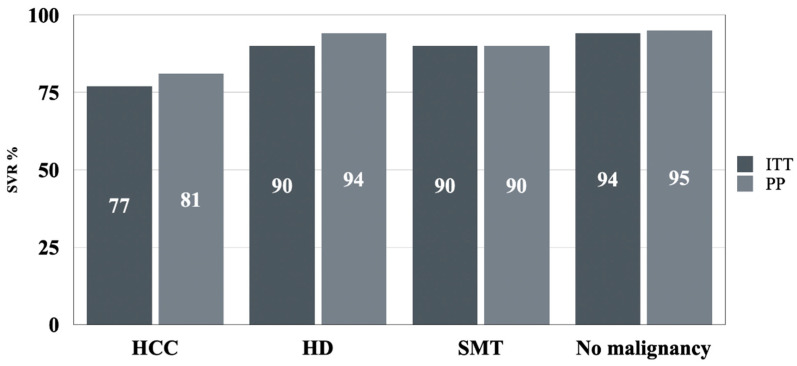
Efficacy of DAAs in patients with active oncological disease: SVR rates in ITT and PP analyses: excluding non-virologic failures. HCC, hepatocellular carcinoma; HD, hemato-oncologic disease; ITT, intention to treat; PP, per protocol; SMT, solid malignant tumor; SVR, sustained virologic response.

**Figure 3 cancers-16-03114-f003:**
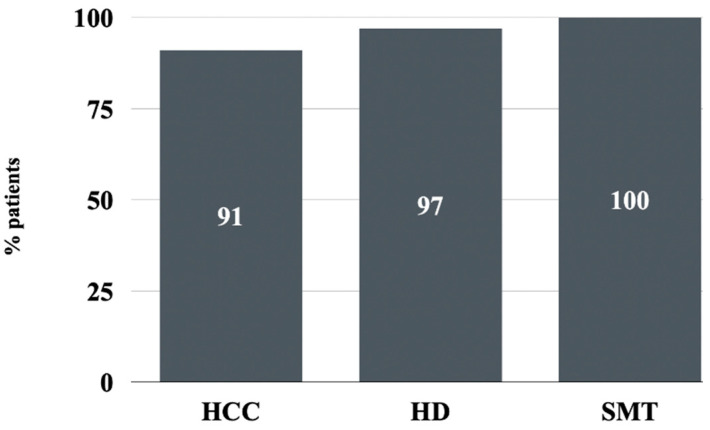
Percentage of patients completing DAA therapy as recommended. DAA, direct-acting antiviral; HCC, hepatocellular carcinoma; HD, hemato-oncologic disease; SMT, solid malignant tumor.

**Table 1 cancers-16-03114-t001:** Baseline characteristics of the investigated groups in comparison to the patients without any malignancy.

Parameter	HCC *n* = 74	HD *n* = 67	SMT *n* = 62	NAM *n* = 12,983	*p*=
Gender, males/females, *n* (%)	47 (63.5)/27 (36.5)	44 (65.7)/23 (34.3)	29 (46.8)/35 (53.2)	6323 (48.7)/6656 (51.3)	0.002
Age (years) mean, SD Median (Q1–Q3)	62, 10.1 62 (57–68)	59.3, 14.6 59 (49.5–70)	60.2, 12.2 62 (52–68)	51.5, 25.1 52 (39–62)	<0.001
BMI median (Q1–Q3)	25.9 (22.1–27)	25 (23.1–29)	25.7 (22.5–30)	25.8 (23.1–29)	0.48
Any comorbidity, *n* (%)	49 (66.22)	31 (36.27)	28 (43.75)	5054 (38.94)	<0.001
Arterial hypertension, *n* (%)	40 (54.05)	23 (34.33)	27 (42.19)	4165 (32.09)	<0.001
Diabetes, *n* (%)	20 (27.03)	11 (16.42)	8 (12.5)	1491 (11.49)	<0.001
Autoimmune disease, *n* (%)	4 (5.41)	2 (2.99)	2 (3.13)	619 (4.77)	0.82
Kidney disease, *n* (%)	5 (6.76)	5 (7.46)	5 (7.81)	501 (3.86)	0.09
Depression, *n* (%)	3 (4.05)	1 (1.49)	4 (6.26)	487 (3.75)	0.56
ALT IU/L median (Q1–Q3)	81 (51–134)	52 (38–101)	59.5 (37.5–95)	58.0 (37–97)	<0.001
Bilirubin mg/dL median (Q1–Q3)	1 (0.72–1.4)	1 (0.52–1.15)	0.6 (0.46–0.8)	0.6 (0.47–0.91)	<0.001
Albumin g/dL median (Q1–Q3)	3.7 (3.2–4.1)	4 (3.4–4.18)	4.0 (3.7–4.18)	4.1 (3.8–4.4)	<0.001
Hemoglobin g/dL Median (Q1–Q3)	13.7 (12.4–14.7)	13 (12.4–14.3)	13.9 (12.7–14.7)	14.5 (13.4–15.5)	<0.001
Platelets ×1000/μL, median (Q1–Q3)	125 (76–165)	125 (87–186)	187.5 (153–249)	194.0 (143–240)	<0.001
HCV RNA ×10^6^ IU/mL, median (Q1–Q3)	664,500 (160,500–1,518,281.5)	1,680,000 (337,376–4,900,000)	541,000 (239,000–1,820,000)	949,118 (317,000–2,463,982.5)	<0.001

**Table 2 cancers-16-03114-t002:** Liver disease characteristics in investigated populations.

Parameter	HCC *n* = 74	HD *n* = 67	SMT *n* = 62	NAM *n* = 12,983	*p*=
Liver fibrosis, *n* (%)					<0.001
F0	0 (0.0)	1 (1.5)	0 (0.0)	270 (2.1)	
F1	3 (4.0)	14 (21.0)	16 (25.8)	5107 (39.3)	
F2	4 (5.4)	16 (23.9)	16 (25.8)	2444 (18.8)	
F3	3 (4.0)	16 (23.9)	14 (22.6)	1812 (13.9)	
F4	62 (83.8)	19 (28.3)	14 (22.6)	3115 (24.0)	
no data	2 (2.7)	1 (1.5)	2 (3.2)	235 (1.8)	
Documented esophageal varices, *n* (%)	30 (40.5)	5 (7.5)	5 (8.1)	974 (7.5)	<0.001
No data	11 (14.9)	12 (17.9)	14 (22.6)	2907 (22.4)	
Ascites at baseline, *n* (%)	6 (8.1)	2 (3.0)	1 (1.6)	151 (1.2)	<0.001
Encephalopathy at baseline, *n* (%)	3 (4.0)	1 (1.5)	1 (1.6)	70 (0.5)	0.2
Child–Pugh, *n* (%)					<0.001
B	11 (14.9)	9 (13.4)	2 (3.2)	354 (2.7)	
C	1 (1.3)	0 (0.0)	0 (0.0)	17 (0.1)	
HBV-coinfection, *n* (%)	13 (17.6)	10 (14.9)	10 (16.1)	1689 (13.0)	0.6
HIV-coinfection *n* (%)	2 (2.7)	2 (3.0)	1 (1.6)	729 (5.6)	0.17

**Table 3 cancers-16-03114-t003:** Treatment effectiveness according to the malignancy presence.

	HCC	HD	SMT	NAM	*p*=
ITT, *n* (%)	57/74 (77)	60/67 (89.6)	56/62 (90)	12,242/12,919 (94.8)	<0.001
PP, *n* (%)	54/67 (80.6)	59/63 (93.6)	55/61 (90.2)	12,100/12,684 (95.4)	<0.001

**Table 4 cancers-16-03114-t004:** Characteristics of treatment discontinuations in HCC patients.

Patient	Age	F, CP	Regimen	History of Previous Therapy	Baseline HCV RNA	Treatment Course Discontinued in	Reason for Discontinuation	Comorbidities	Anticancer Concomitant Medications
Male 1	76	4, no data	LDV/SOF + RBV, 12 weeks	relapser	708150	week 4	performance status deterioration	hypertension, diabetes	sorafenib
Male 2	63	2, A	OBV/PTV/r + DSV + RBV, 24 weeks	treatment-naive	1410000	week 12	HCC progression	hypertension, diabetes	no
Male 3	57	3, A	LDV/SOF + RBV, 24 weeks	Non-responder	615915	week 20	decompensation	kidney disease	no
Male 4	53	4, C	LDV/SOF + RBV, 24 weeks	treatment-naive	50412	week 20	HCC diagnosed during treatment	no data	no
Male 5	60	4, B	OBV/PTV/r + DSV, 12 weeks	treatment-naive	103000	week 8	decompensation	hypertension, porphyria	no
Male 6	65	4, A	OBV/PTV/r + RBV, 24 weeks	Non-responder	3310000	week 12	liver transplantation	hypertension	no
Female 1	60	4, B	OBV/PTV/r + DSV, 12 weeks	treatment-naive	81200	week 5	decompensation	diabetes, atherosclerosis	sorafenib

F, fibrosis; CP, Child–Pugh scale; HCV RNA, ribonucleic acid of hepatitis C virus; LDV, lamivudine; SOF, sofosbuvir; RBV, ribavirin; OBV, ombitasvir; PTV, paritaprevir; r, ritonavir; DSV, dasabuvir.

**Table 5 cancers-16-03114-t005:** Characteristics of treatment discontinuations in HD patients.

Patient	Age	Malignancy	F, CP	Regimen	History of Previous Therapy	Baseline HCV RNA	Treatment Course Discontinued in	Reason for Discontinuation	Comorbidities	Anticancer Concomitant Medications
Female 1	40	multiple myeloma	4, A	LDV/SOF, 12 weeks	treatment-naive	5167178	week 20	myeloma exacerbation	no	Bortezomib
Male 1	80	Non-Hodgkin Lymphoma	3, B	VEL/SOF, 12 weeks	treatment-naive	14900000	week 2	death	hypertension, diabetes	no

F, fibrosis; CP, Child–Pugh scale; HCV RNA, ribonucleic acid of hepatitis C virus; LDV, lamivudine; SOF, sofosbuvir; VEL, velpatasvir.

## Data Availability

The data supporting reported results can be provided upon request from the corresponding author.

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
