# Peer review of "The Real-World Efficacy and Safety of Direct-Acting Antivirals for Chronic Hepatitis C in Patients Active Malignancies"

_cancers, 2024, doi:10.3390/cancers16173114_

Round 1

Reviewer 1 Report

Comments and Suggestions for Authors

Interesting study. The authors could provide some data on the kinetics of liver fibrosis in patients achieving SVR. If the authors do not have these data, at least they should comment this aspect in the discussion 8cite the recent series PMID: 28976021).

The number of references is too low and most of them are not updated

Author Response

Reviewer 1

Interesting study. The authors could provide some data on the kinetics of liver fibrosis in patients achieving SVR. If the authors do not have these data, at least they should comment this aspect in the discussion 8cite the recent series PMID: 28976021).

The number of references is too low and most of them are not updated

Thank you for your comments which allowed to improve the manuscript. All comments were included in the most recent version  of the manuscript and highlighted in yellow. Additional references were included. Currently there are 30 references. Suggested reference (28976021) was also included.

Reviewer 2 Report

Comments and Suggestions for Authors

Dabrowska et al. reported that the efficacy and safety of direct-acting antivirals for chronic hepatitis C in patients with hepatocellular carcinoma and other malignancies in a large real-world study.

1.    In simple summary, authors wrote the results and conclusion clearly.

2.    In abstract section, “We claim high effectiveness and good safety profile of the DAA in patients with active SMT and HD, while premature therapy discontinuations due to the liver disease progression being major concern for patients with HCC.” Need new knowledge and compare with the previous results.

3.    In lines 51, make changes from (17,2 % vs. SMT=10,3%, HCC=8,2%, without=7,8%, p=0.04) to (17.2 % vs. SMT=10.3%, HCC=8.2%, without=7.8%, p=0.04). In text, tables and figures, please fix similar things.

4.    In lines 92-95, “Therefore, it is crucial that potential chronic HCV infections are consistently detected and treated in patients suffering from hepatocellular carcinoma, solid malignancies or hematological ones. This might require elaboration of a novel systemic approach in the overall cancer treatment planning.” Use HCC, SMT, and HD.

5.    In lines 100-102, use HCC, SMT, and HD.

6.    In lines 142-144, use HCC, SMT, and HD.

7.    In Table 1, use “Age” and “Gender.” Please fix all misspellings.

8.    Add references and discuss more: Kanda, T.; Matsumoto, N.; Ishii, T.; Arima, S.; Shibuya, S.; Honda, M.; Sasaki-Tanaka, R.; Masuzaki, R.; Kanezawa, S.; Nishizawa, T.; et al. Chronic Hepatitis C: Acute Exacerbation and Alanine Aminotransferase Flare. Viruses 2023, 15, 183. https://doi.org/10.3390/v15010183

Comments on the Quality of English Language

Dabrowska et al. reported that the efficacy and safety of direct-acting antivirals for chronic hepatitis C in patients with hepatocellular carcinoma and other malignancies in a large real-world study.

1.    In simple summary, authors wrote the results and conclusion clearly.

2.    In abstract section, “We claim high effectiveness and good safety profile of the DAA in patients with active SMT and HD, while premature therapy discontinuations due to the liver disease progression being major concern for patients with HCC.” Need new knowledge and compare with the previous results.

3.    In lines 51, make changes from (17,2 % vs. SMT=10,3%, HCC=8,2%, without=7,8%, p=0.04) to (17.2 % vs. SMT=10.3%, HCC=8.2%, without=7.8%, p=0.04). In text, tables and figures, please fix similar things.

4.    In lines 92-95, “Therefore, it is crucial that potential chronic HCV infections are consistently detected and treated in patients suffering from hepatocellular carcinoma, solid malignancies or hematological ones. This might require elaboration of a novel systemic approach in the overall cancer treatment planning.” Use HCC, SMT, and HD.

5.    In lines 100-102, use HCC, SMT, and HD.

6.    In lines 142-144, use HCC, SMT, and HD.

7.    In Table 1, use “Age” and “Gender.” Please fix all misspellings.

8.    Add references and discuss more: Kanda, T.; Matsumoto, N.; Ishii, T.; Arima, S.; Shibuya, S.; Honda, M.; Sasaki-Tanaka, R.; Masuzaki, R.; Kanezawa, S.; Nishizawa, T.; et al. Chronic Hepatitis C: Acute Exacerbation and Alanine Aminotransferase Flare. Viruses 2023, 15, 183. https://doi.org/10.3390/v15010183

Author Response

Thank you for your comments which allowed to improve the manuscript. All comments were included in the most recent version  of the manuscript and highlighted in yellow. Specific answers are listed below:

.    In simple summary, authors wrote the results and conclusion clearly.

  1. In abstract section, “We claim high effectiveness and good safety profile of the DAA in patients with active SMT and HD, while premature therapy discontinuations due to the liver disease progression being major concern for patients with HCC.” Need new knowledge and compare with the previous results.
    Answer: The discussions were substantially expanded and new data and references added.
  2. In lines 51, make changes from (17,2 % vs. SMT=10,3%, HCC=8,2%, without=7,8%, p=0.04) to (17.2 % vs. SMT=10.3%, HCC=8.2%, without=7.8%, p=0.04). In text, tables and figures, please fix similar things.
    Answer: This line was corrected according to reviewer suggestion
  3. In lines 92-95, “Therefore, it is crucial that potential chronic HCV infections are consistently detected and treated in patients suffering from hepatocellular carcinoma, solid malignancies or hematological ones. This might require elaboration of a novel systemic approach in the overall cancer treatment planning.” Use HCC, SMT, and HD.
    Answer: This sentence was corrected according to reviewer suggestion
  4. In lines 100-102, use HCC, SMT, and HD.
    Answer: This sentence was corrected according to reviewer suggestion
  5. In lines 142-144, use HCC, SMT, and HD.
    Answer: This sentence was corrected according to reviewer suggestion
  6. In Table 1, use “Age” and “Gender.” Please fix all misspellings.
    Answer: Requested corrections were done.
  7. Add references and discuss more: Kanda, T.; Matsumoto, N.; Ishii, T.; Arima, S.; Shibuya, S.; Honda, M.; Sasaki-Tanaka, R.; Masuzaki, R.; Kanezawa, S.; Nishizawa, T.; et al. Chronic Hepatitis C: Acute Exacerbation and Alanine Aminotransferase Flare. Viruses 2023, 15, 183. https://doi.org/10.3390/v15010183
    Answer: Mentioned above reference was added

Reviewer 3 Report

Comments and Suggestions for Authors

I would like to congratulate the authors on the work they have done. This is an important study that provides very interesting information on direct-acting antiviral treatments in patients with various neoplastic processes. It offers valuable insights into the efficacy and safety of these treatments in such patients.

The methodology is sound. The structure is well-organized, and it includes all the necessary content.

The results are well-explained and correlated with published studies. The references are current and appropriate. The tables and figures are also suitable.

I would only suggest reconsidering the title of the article. It seems excessively long. A good title should contain no more than 16-18 words, and I believe it should be adjusted accordingly.

Author Response

I would like to congratulate the authors on the work they have done. This is an important study that provides very interesting information on direct-acting antiviral treatments in patients with various neoplastic processes. It offers valuable insights into the efficacy and safety of these treatments in such patients.

The methodology is sound. The structure is well-organized, and it includes all the necessary content.

The results are well-explained and correlated with published studies. The references are current and appropriate. The tables and figures are also suitable.

I would only suggest reconsidering the title of the article. It seems excessively long. A good title should contain no more than 16-18 words, and I believe it should be adjusted accordingly.

Answer:

Thank you for your comments which allowed to improve the manuscript. The title was simplified according to reviewer suggestion

Round 2

Reviewer 1 Report

Comments and Suggestions for Authors

The revised manuscript is OK. Thank you!

Reviewer 2 Report

Comments and Suggestions for Authors

Of Reference 12, take out "12".